

# Evaluation of nutritional and economic feed values of spent coffee grounds and *Artemisia princeps* residues as a ruminant feed using *in vitro* ruminal fermentation

Jakyeom Seo[1,2], Jae Keun Jung[1] and Seongwon Seo[1]

[1] Animal Biosystem Sciences, Chungnam National University, Daejeon, Republic of Korea
[2] Life and Industry Convergence Research Institute, Department of Animal Science, Pusan National University, Miryang, South Korea

## ABSTRACT

Much research on animal feed has focused on finding alternative feed ingredients that can replace conventional ones (e.g., grains and beans) to reduce feed costs. The objective of this study was to evaluate the economic, as well as nutritional value of spent coffee grounds (SCG) and Japanese mugwort (*Artemisia princeps*) residues (APR) as alternative feed ingredients for ruminants. We also investigated whether pre-fermentation using *Lactobacillus* spp. was a feasible way to increase the feed value of these by-products. Chemical analyses and an *in vitro* study were conducted for SCG, APR, and their pre-fermented forms. All the experimental diets for *in vitro* ruminal fermentation were formulated to contain a similar composition of crude protein, neutral detergent fiber and total digestible nutrients at 1x maintenance feed intake based on the dairy National Research Council (NRC). The control diet was composed of ryegrass, corn, soybean meal, whereas the treatments consisted of SCG, SCG fermented with *Lactobacillus* spp. (FSCG), APR, and its fermented form (FAPR). The treatment diets replaced 100 g/kg dry matter (DM) of the feed ingredients in the control. Costs were lower for the all treatments, except FAPR, than that of the control. After 24-h incubation, the NDF digestibility of the diets containing SCG and its fermented form were significantly lower than those of the other diets ($P < 0.01$); pre-fermentation tended to increase NDF digestibility ($P = 0.07$), especially for APR. Supplementation of SCG significantly decreased total gas production (ml/g DM) after 24-h fermentation in comparison with the control ($P < 0.05$); however, there were no significant differences between the control and the SCG or the APR diets in total gas production, as expressed per Korean Won (KRW). Diets supplemented with SCG or FSCG tended to have a higher total volatile fatty acid (VFA) concentration, expressed as per KRW, compared with the control ($P = 0.06$). Conversely, the fermentation process of SCG and APR significantly decreased total gas production and VFA production as expressed per KRW ($P < 0.05$). Because of their nutrient composition and relatively lower cost, we concluded that SCG and APR could be used as alternative feed sources, replacing conventional feed ingredients. However, pre-fermentation of agricultural by-products, such as SCG and APR, may be inappropriate for improving their nutritive considering the increase in production costs.

Corresponding author
Seongwon Seo, swseo@cnu.ac.kr

# INTRODUCTION

Research on animal feed has often focused on finding alternative feed ingredients to replace conventional ones (e.g., grains and beans) in order to reduce feed costs. This is most important in developing countries where the supplies of cereal grains and beans are not great enough to support even the human population. Historically, by-products from processing crops and food products have received much attention as feed alternatives because of their consistent and mass production. Food by-products would also likely be inexpensive because of their classification as a waste product. Many by-products, however, do not contain enough nutrients to support livestock requirements, and their palatability and digestibility would need to be enhanced, even for ruminants. Pre-fermentation of these by-products by bacteria (*Han, 1975*; *Han, 1978*), yeast (*Wanapat et al., 2011*), or fungi (*Salman et al., 2008*) are common methods to enhance their nutritional value (see *Mahesh & Mohini (2013)* for a review).

Although fermentation may improve the nutritional quality of the by-products, and it is an environmentally sustainable practice, it will increase the cost for feed production. This additional cost is often ignored by researchers. To make the fermentation process feasible, the production costs of a fermented by-product must to be competitive with conventional feed ingredients (*Wanapat, Kang & Polyorach, 2013*). Therefore, the economic value, as well as nutritional value, of fermented by-products must be considered simultaneously.

In this study, we evaluated the economic and nutritional values of spent coffee grounds (SCG) and Japanese mugwort (*Artemisia princeps*) residues (APR), as well as their fermented products as potentially cost-effective feed ingredients for ruminants. An *in vitro* ruminal fermentation study where Total mixed rations (TMR) was supplemented with SCG or APR was conducted. These by-products were chosen primarily because of the rapid increase of production, and the relatively high content of nutrients and bio-active compounds. SCG are generated during the manufacture of instant coffee. It is the residue that remains after brewing raw coffee powders with hot water or steam. Annually, 6-million tons of SCG are produced worldwide and most of it is burned as waste, resulting in greenhouse gas emissions (*Tokimoto et al., 2005*). There have been studies on the potential of SCG as a feed source for ruminants (*Campbell et al., 1976*; *Bartley et al., 1978*; *Givens & Barber, 1986*; *Xu et al., 2007*) and even for monogastric animals (*Sikka, Bakshi & Ichhponani, 1985*; *Sikka & Chawla, 1986*). *Xu et al. (2007)* concluded that wet coffee grounds could be included up to 100 g/kg on a dry matter (DM) basis in total mixed rations (TMR) for goats. APR is a by-product from the traditional Korean medicine industry; it is produced during harvesting and processing of leaves. Positive effects were observed in growth performance of broilers supplemented by *Lactobacillus*-fermented APR (*Kim et al., 2012*). To the best of our knowledge, however, there has been no attempt to assess the feasibility of fermented SCG or APR as a feed alternative in ruminants based on both their nutritional and economic values.

## MATERIALS AND METHODS

Two cannulated non-lactating Holstein cows at Center for Animal Science Research, Chungnam National University, Korea were used in this study. Animal use and the protocols for this experiment were reviewed and approved by the Chungnam National University Animal Research Ethics Committee (CNU-00455).

### Preparation of experimental diets

The feed ingredients used in this study were ryegrass, corn, soy bean meal (SBM), SCG, APR, and the *Lactobacillus*-fermented forms of SCG and APR (FSCG and FAPR, respectively). SCG were purchased from an instant coffee manufacturer (Dongseo Food, Inc., Bupyeong, Korea), and APR was obtained from the Ganghwa Agricultural R&D Center (Incheon, Korea). The fermentation process for SCG and APR with *Lactobacillus* spp. was conducted as described by *Kim et al. (2012)*. Briefly, four strains of *Lactobacillus* spp. (*L. acidophilus* ATCC 496, *L. fermentum* ATCC 1493 (American Type Culture Collection: Virginia, USA), *L. plantarum* KCTC 1048 (Korean Collection Type Culture, Daejeon, Korea), and *L. casei* IFO 3533 (Korea Food Research Institute, Daejeon, Korea)) were used to ferment SCG and APR. A 1l culture medium was inoculated with a 2 ml aliquot containing $10^9$ cfu/ml of each *Lactobacillus* strain (de Man, Rogosa, and Sharpe broth (Difco Laboratories, Francisco Soria Melquizo S.A., Madrid, Spain), 10 g; sucrose, 10 g) and incubated at 36 °C for 24 h (h). Next, 4 kg of dried SCG and APR were mixed with 400 ml of prepared *Lactobacillus* inoculum in a fermentation flask and incubated for 72 h at 36 °C. The fermented by-products were obtained after freeze-drying the cultured substrate and media for 2 days following the manufacture's recommendations (ilShinBioBase, Inc., Korea).

All of the diets that included *in vitro* fermentation were formulated to meet nutrient requirements for non-lactating dairy cows (total digestible nutrient at 1x maintenance feed intake (TDN1x), 680 g/kg; crude protein (CP), 120 g/kg; neutral detergent fiber (NDF), 420 g/kg on a DM basis) according to the National Research Council (*NRC, 2001*). The control diet was composed of 500 g/kg of ryegrass and 500 g/kg of a corn and SBM mix. The four experimental diets (SCG, FSCG, APR, and FAPR) were formulated to contain TDN1x, CP, and NDF contents (g/kg DM) similar to the control, replacing original ingredients with 100 g/kg DM of SCG, FSCG, APR or FAPR, respectively.

### *In vitro* incubation

Rumen fluid was collected before the morning feeding from two cannulated non-lactating Holstein cows fed a ration consisting of 600 g/kg timothy hay and 400 g/kg of a commercial concentrate mix ($123 \pm 8.8$ g/kg CP, $35 \pm 6.4$ g/kg ether extract (EE), $265 \pm 6.9$ g/kg NDF, and $109 \pm 1.2$ g/kg ash) twice daily at the Center for Animal Science Research, Chungnam National University, Korea. The rumen contents were mixed and transferred into a thermos bottle, and immediately transported to the laboratory. Rumen contents were strained through 4 layers of cheesecloth and mixed with 4x volumes of *in vitro* rumen buffer solution (*Goering & Van Soest, 1970*) under strictly anaerobic conditions. Fifty ml

of rumen fluid/buffer mixture was transferred into 125 ml serum bottles containing 0.5 g of experimental diets under continuous flushing with $O_2$-free $CO_2$ gas. The bottles were sealed with butyl rubber stoppers and aluminum caps, and incubated for 0, 3, 6, 9, 12, 24, and 48 h at 39 °C.

## Analyses

Contents of DM (#934.01), CP (#976.05), EE (#920.39), lignin (ADL; #973.18) and ash (#942.05) in the feed samples were determined as described by *AOAC (2005)*. NDF, analyzed using a heat stable amylase and expressed inclusive of residual ash, and acid detergent fiber (ADF) were determined as described by *Van Soest, Robertson & Lewis (1991)*. Neutral detergent insoluble crude protein (NDICP) and acid detergent insoluble crude protein (ADICP) were determined as described by *Licitra, Hernandez & Van Soest (1996)*. Non-fiber carbohydrate (NFC) was calculated as 1000-CP-EE-Ash-(NDF-NDICP) based on *NRC (2001)*.

For phenolic acid analysis, caffeic acid, *p*-coumaric acid, ferulic acid, and sinapic acid (#C0625, #C9008, #12870, and #D7927, respectively; Sigma-Aldrich, St. Louis, Missouri, USA) were used as standards. Identification of phenolic acids in the SCG and APR samples was based primarily on retention time, UV spectra obtained from high-performance liquid chromatography (HPLC-DAD), and mass spectrometric data using authentic standards. Phenolic acids were analyzed as described by *Rochfort et al. (2006)*.

After each incubation period, total gas production was measured using a pressure transducer (Sun Bee Instruments, Inc., Seoul, Korea) as described by *Theodorou et al. (1994)*. Next, 5 ml of head space gas was collected using a gas tight syringe (Hamilton, Reno, Nevada, USA) for analysis of $CH_4$ using a gas chromatograph (Daesung Science IGC-7200, Seoul, Korea) equipped with a thermal conductivity detector and HayeSep Q 80/100 column (Restek, Bellefonte, Pennsylvania, US). The pH of the cultured fluid was measured with a general purpose pH meter (Istek Inc., Seoul, Korea). The cultured fluid was then centrifuged at 14,000 rpm for 10 min at 4 °C, and the supernatant was used for the analyses of volatile fatty acid (VFA) and ammonia concentrations. The remaining undegraded samples and fluid were analyzed for NDF using a modified version of the micro-NDF method proposed by *Pell & Schofield (1993)* for measuring NDF degradability. VFA concentrations were determined as described by *Erwin, Marco & Emery (1961)*. Ammonia concentration was analyzed by the method of *Chaney & Marbach (1962)*.

For the economic analysis, the price of each feed ingredient was estimated and expressed in Korean won (KRW). The average market prices of ryegrass, corn, and SBM in 2009 were used (i.e., 364, 317, and 660 KRW/kg DM, respectively). The prices of SCG and APR (i.e., 91 and 235 KRW/kg DM, respectively) were obtained from the respective manufacturers. Prices of FSCG and FAPR (i.e., 270 and 429 KRW/kg DM, respectively) were calculated by adding SCG and APR prices to the cost for the fermentation process (i.e., 200 KRW/kg as fed). The cost of the fermentation process included expenses for microbial strains, culture medium, labor, use of fermenter instruments, and manufacturing.

**Table 1 Chemical composition (g/kg, DM basis) and estimated price (KRW/kg DM) of each feed ingredient.**

| Item | Feed ingredients[a] | | | | | | |
|---|---|---|---|---|---|---|---|
| | Ryegrass | Corn | SBM | SCG | FSCG | APR | FAPR |
| DM (g/kg as fed) | 954 | 907 | 900 | 550 | 925 | 850 | 932 |
| CP | 81 | 103 | 513 | 138 | 141 | 107 | 103 |
| EE | 8 | 34 | 15 | 136 | 157 | 18 | 17 |
| Ash | 54 | 42 | 62 | 20 | 18 | 91 | 95 |
| NDF | 739 | 116 | 237 | 656 | 792 | 649 | 706 |
| ADF | 481 | 61 | 105 | 451 | 526 | 538 | 523 |
| ADL | 54 | 22 | 26 | 142 | 146 | 99 | 74 |
| NDICP | 46 | 33 | 217 | 105 | 114 | 59 | 47 |
| ADICP | 22 | 32 | 139 | 95 | 100 | 42 | 23 |
| NFC | 164 | 738 | 390 | 155 | 6 | 194 | 127 |
| Feed price[b] (KRW/kg DM) | 364 | 317 | 660 | 91 | 270 | 235 | 429 |

**Notes.**

[a] SBM; soybean meal, SCG; spent coffee grounds, FSCG; fermented spent coffee grounds, APR; *Artemisia princeps* residues, FAPR; fermented *Artemisia princeps* residues.

[b] The prices for ryegrass, corn, and SBM are the average market prices in Korea in 2009. The prices of SCG and APR were obtained from the respective manufacturers (Dongseo Food, Inc., and Ganghwa Agricultural R&D Center, respectively). For the prices of FSCG and FAPR, the cost for the fermentation process, including the expenses for microbial strains, culture medium, labor, and use of fermenter instruments was calculated and added to their raw prices.

## Statistical analysis

The experiment was conducted using a completely randomized design, and the data were analyzed using the GLM procedure of SAS (SAS Institute Inc., Carey, North Carolina, USA) as:

$$y_{ij} = \mu + \tau_i + e_{ij}$$

where: $y_{ij}$ is the $j$th observation in the $i$th treatment, $\mu$ is the overall mean, $\tau_i$ is the fixed effect of the $i$th treatment ($i = 1$–$5$), and $e_{ij}$ is the unexplained random effect on the $j$th observation in the $i$th treatment. Four contrasts were tested: the difference between the control and SCG (control *versus* SCG and FSCG), control and APR (control *versus* APR and FAPR), SCG and APR (SCG and FSCG *versus* APR and FAPR), and non-fermented and fermented groups (SCG and APR *versus* FSCG and FAPR). Differences among treatments were also compared with the Tukey's test when there was a significant overall treatment effect. Statistical significance was defined as $P < 0.05$, and a trend was discussed at $0.05 \leq P < 0.10$.

## RESULTS

Both by-products had higher amounts of fiber and lower NFC compared with corn or SBM, and SCG had a markedly high level ($>100$ g/kg DM) of EE (Table 1). Phenolic acid contents of SCG and APR were 3.22 and 2.80 mg/g DM, respectively. More specifically, the concentrations of caffeic acid, *p*-coumaric acid, ferulic acid, and sinapic acid were 1.07, 0.42, 1.70, and 0.04 mg/g DM and 1.72, 0.77, 0.32, and 0.00 mg/g DM in SCG and APR,

**Table 2** Ingredients, analyzed chemical composition (g/kg, DM basis), and estimated cost (KRW/kg DM) of experimental diets.

| Items | Experimental diets[b] | | | | |
| --- | --- | --- | --- | --- | --- |
| | Control | SCG | FSCG | APR | FAPR |
| Ingredients[a] | | | | | |
| Ryegrass | 500 | 430 | 408 | 393 | 386 |
| Corn | 432 | 414 | 438 | 445 | 451 |
| SBM | 69 | 56 | 54 | 62 | 63 |
| SCG | 0 | 100 | 0 | 0 | 0 |
| FSCG | 0 | 0 | 100 | 0 | 0 |
| APR | 0 | 0 | 0 | 100 | 0 |
| FAPR | 0 | 0 | 0 | 0 | 100 |
| Analyzed chemical composition | | | | | |
| DM | 930 | 929 | 933 | 933 | 926 |
| CP | 120 | 120 | 120 | 120 | 120 |
| EE | 20 | 32 | 35 | 21 | 21 |
| Ash | 50 | 46 | 46 | 53 | 53 |
| NDF | 436 | 444 | 444 | 423 | 432 |
| ADF | 274 | 283 | 281 | 277 | 272 |
| ADL | 38 | 48 | 48 | 42 | 40 |
| NDICP | 52 | 56 | 57 | 52 | 51 |
| ADICP | 34 | 40 | 40 | 36 | 34 |
| NFC | 427 | 414 | 412 | 435 | 425 |
| TDN1x | 680 | 683 | 687 | 677 | 680 |
| Estimated cost (KRW/kg DM) | 364 | 334 | 350 | 348 | 368 |

**Notes.**

[a] SBM; soybean meal, SCG; spent coffee grounds, FSCG; fermented spent coffee grounds, APR; *Artemisia princeps* residues, FAPR; fermented *Artemisia princeps* residues.

[b] The SCG, FSCG, APR and FAPR treatments contained 100 g/kg of SCG, FSCG, APR, and FAPR, respectively.

respectively. The fermentation process increased feed cost and decreased NFC contents in both by-products (Table 1). The fermentation process also increased NDF content.

All the experimental diets for the *in vitro* fermentation contained similar TDN1x (677–683 g/kg DM), CP (120 g/kg DM), and NDF (423–444 g/kg DM) contents (Table 2), and their prices were calculated based on the prices of each feed ingredient (Table 1) and the formulated ratio of respective treatments (Table 2). The Korea Won was the unit to estimate cost of experimental diets. Compared to the control diet, inclusion of the by-products reduced the price of formulated diet (14–30 KRW/kg DM), except for FAPR which increased the price by 4 KRW/kg DM (Table 2). The use of fermented by-products increased the cost of diets by 6–20 KRW/kg DM.

Supplementation of APR or FAPR significantly reduced pH compared to other treatments ($P < 0.01$, Table 3). Ammonia nitrogen concentration did not differ among treatments. NDF digestibility of the SCG group was significantly lower than that of the other diets ($P < 0.01$) and the fermentation process tended to increase NDF digestibility ($P = 0.07$), especially that of APR ($P < 0.05$, Table 3). Compared to the control diet,

**Table 3** Fermentation characteristics and $CH_4$ production after 24 h *in vitro* incubation of the experimental diets using strained ruminal fluid.

| Item | Treatments[a] | | | | | SEM[b] | P-value | | | | |
|---|---|---|---|---|---|---|---|---|---|---|---|
| | Control | SCG | FSCG | APR | FAPR | | Overall | Control vs. SCG | Control vs. APR | SCG vs. APR | Fermentation[c] |
| pH | 6.58[d] | 6.59[d] | 6.58[d] | 6.55[e] | 6.55[e] | 0.000 | <0.01 | 0.07 | <0.01 | <0.01 | 0.63 |
| $NH_3$–N, mg/100 ml | 11.9 | 12.1 | 11.5 | 11.7 | 11.5 | 0.24 | 0.39 | 0.79 | 0.27 | 0.30 | 0.20 |
| NDF digestibility, g/kg DM | 293.5[de] | 252.4[e] | 242.3[e] | 259.8[e] | 316.9[d] | 11.43 | <0.01 | <0.01 | 0.72 | <0.01 | 0.07 |
| Gas production, ml/g DM | 193.7 | 184.6 | 180.7 | 192.2 | 184.4 | 3.20 | 0.07 | <0.05 | 0.20 | 0.11 | 0.10 |
| Gas production, ml/KRW | 532.7[de] | 553.5[d] | 516.9[de] | 551.8[d] | 501.9[e] | 9.03 | <0.01 | 0.83 | 0.61 | 0.38 | <0.01 |
| $CH_4$, ml/g DM | 19.2 | 18.1 | 18.0 | 18.8 | 20.3 | 0.78 | 0.27 | 0.24 | 0.71 | 0.07 | 0.39 |
| VFA profiles | | | | | | | | | | | |
| Total VFA, mmol/g DM | 5.99 | 5.87 | 5.82 | 5.97 | 6.06 | 0.090 | 0.41 | 0.24 | 0.80 | 0.09 | 0.80 |
| Total VFA, mmol/KRW | 16.43 | 17.60 | 16.67 | 17.13 | 16.50 | 0.267 | 0.05 | 0.06 | 0.27 | 0.26 | <0.05 |
| Acetate, mmol/mol | 602.0 | 601.0 | 598.9 | 599.7 | 599.5 | 1.64 | 0.67 | 0.32 | 0.25 | 0.84 | 0.51 |
| Propionate, mmol/mol | 201.2 | 199.4 | 198.4 | 201.0 | 200.9 | 1.37 | 0.55 | 0.20 | 0.88 | 0.16 | 0.70 |
| Butyrate, mmol/mol | 166.6[e] | 168.8[de] | 172.1[d] | 168.8[de] | 169.1[de] | 0.80 | 0.01 | <0.01 | <0.05 | 0.08 | <0.05 |
| A/P ratio | 2.99 | 3.02 | 3.02 | 2.98 | 2.98 | 0.028 | 0.81 | 0.43 | 0.89 | 0.26 | 0.91 |

**Notes.**

[a] SCG; spent coffee grounds, FSCG; fermented spent coffee grounds, APR; *Artemisia princeps* residues, FAPR; fermented *Artemisia princeps* residues. The SCG, FSCG, APR and FAPR treatments contained 100 g/kg of SCG, FSCG, APR, and FAPR, respectively.

[b] SEM; standard error of the mean.

[c] Statistical difference between fermented and non-fermented substrates (SCG, APR vs. FSCG, FAPR).

[d,e] Means that do not have common superscript differ significantly ($P < 0.05$).

supplementation of SCG significantly decreased total gas production (ml/g DM) after 24 h fermentation ($P < 0.05$, Table 3). The fermentation process of SCG and APR tended to decrease total gas production ($P = 0.10$) and significantly decreased total gas production per KRW (ml/KRW). No statistical difference in $CH_4$ production (ml/g DM) was observed among treatments; however, there was a tendency for a decrease in $CH_4$ production with SCG supplementation as compared to that of APR ($P = 0.07$). Significant differences were not detected in total VFA production (mmol/g DM), proportion of acetate (mmol/mol), proportion of propionate (mmol/mol), or acetate:propionate ratio. The diet supplemented with SCG or FSCG; however, tended to have a higher total VFA concentration per KRW (mmol/KRW) compared to the control ($P = 0.06$). Moreover, the fermentation process of SCG and APR significantly decreased total VFA production (mmol/KRW) ($P < 0.05$). There were significant differences among the experimental diets in the proportion of butyrate ($P < 0.01$, Table 3).

## DISCUSSION

This study was designed to evaluate the nutritional and economic value of two food by-products, SCG and APR, as alternatives to conventional feed ingredients, and to investigate whether the pre-fermentation of these by-products using *Lactobacillus* spp. could be an economically feasible practice for increasing their nutritional value. These by-products were chosen primarily because of the rapid increase in their production, and their relatively high content of nutrients and bio-active compounds (*Wallace, 2007*;

*Acevedo et al., 2013*). It was expected that the bio-active phenolic compounds contained in these by-products would modulate rumen microbial activity, fermentation characteristics, and $CH_4$ emission (*Acevedo et al., 2013*; *Kim et al., 2013*). Fermentation of selected by-products using *Lactobacillus* spp. was applied to improve their nutritive values as feed ingredients. However, a dramatic increase of CP by microbial proliferation was not observed in either fermented by-products (Table 1), indicating that phenolic compounds, such as tannins, appear to be resistant to cell wall degradation by microbial inoculants. The decrease in NFC contents observed in fermented by-products (Table 1) indicated that the inoculated microbes might utilize non fiber sugars in both by-products during fermentation period. Both NDICP and ADICP contents were high in SCG, which might be caused by the Maillard reaction that occurs during beverage production under high temperature (*Senevirathne et al., 2012*).

The use of SCG and APR could decrease the price of diets with similar nutrient contents. Formulated experimental diets using SCG, FSCG, and APR had lower feed cost than the control diet without compromising CP, NDF, or TDN1x contents; however, the experimental diet using FAPR did not (Table 2). This result suggested that the residues (except those of FAPR) may be applied to feed formulation to achieve economic advantages; however, lower NFC and higher EE contents observed in diets having SCG compared to the control diet should be considered when SCG is applied to practical feed formulation. The supplementation fat source can increase energy density in diets but the excessive fat addition limits rumen fermentation, intestinal nutrients absorption (*Palmquist, 1994*). Because of different nutrient compositions between SCG and APR, SCG tends to replace energy and protein sources, whereas APR replaces forage. Thus, supplementation of SCG is more favorable when the price of protein sources is high, whereas APR would be a more appropriate alternative when the price of forage is high. Both SCG and APR are feed ingredient alternatives for ruminants, particularly if the goal is to reduce feed cost.

Total gas and VFA production after 24 h of *in vitro* fermentation were investigated to figure out whether both SCG and APR can be alternative feed ingredients in nutritional and economical aspects. Lower gas production observed in SCG treatments than that of the control could have been caused by lower NFC contents compared to the control diet, although they had similar composition in regard to CP, NDF, and TDN1x. *Seo et al. (2009)* reported that total gas production at 48 h was primarily determined by the NFC content. The diets containing SCG had significantly lower NDF digestibility than the other diets. It was speculated that such phenolic compounds and resistance factors from the Maillard reaction in SCG might be associated with *in vitro* digestibility (*Puchala et al., 2005*; *Senevirathne et al., 2012*). More importantly, there was no significant difference between the control and the SCG supplemented diets in gas production as expressed by ml/KRW, which implied replacement of conventional feed ingredients by SCG could be economically beneficial without compromising ruminal fermentation.

Regarding VFA concentration, diets using APR exhibited a tendency toward increased VFA production at 24 h fermentation compared to SCG (Table 3). Meanwhile, VFA

production based on price (mmol/KRW) tended to be higher in samples using SCG than the control group, indicating the use of SCG as the alternative feed source might not cause any negative effects on rumen fermentation. *Xu et al. (2007)* indicated that the proportion of SCG in TMR should not exceed 10% of DM. Lower nutrient digestibility and VFA production were observed when 20% of SCG was added to TMR in their study. However, since nutrient composition between treatments was not controlled in their studies, the optimal concentration of SCG that could be included in ruminant diets should be tested in terms of both nutritional and economical aspects as described in this study.

Based on the results from *in vitro* ruminal fermentation, the pre-fermentation process in this study is unlikely a feasible practice to improve the nutritional value of SCG and APR. Although pre-fermentation increased NDF digestibility of APR, there was a significant decrease in VFA production in fermented treatments when expressed based on costs (mmol/KRW). A declining tendency in total gas production was also observed in the pre-fermentation of the by-products. This implies that fermented by-products may not be appropriate as feed alternatives in terms of cost-effectiveness, even though they have a potential to be a feed additive having probiotic functions that include modulation of ruminal fermentation, enhanced fiber digestion, and immune stimulation in the hind-gut (*McAllister et al., 2011*).

A previous *in vitro* study using beverage residues found that TMR using SCG had similar $CH_4$ production to the control, whereas, both TMRs showed significantly decreased $CH_4$ production compared to the other TMR using green tea residues (*Senevirathne et al., 2012*). In this study, $CH_4$ production was numerically decreased in the diet formulated with SCG compared to the control; nevertheless, further research to investigate the effect of SCG usage on rumen fermentation is required because it might have considerable amounts of polyphenolic compounds that can modulate $CH_4$ emission (*Puchala et al., 2005*), nitrogen metabolism, and the rumen protozoal population (*Wallace, 2007*).

## CONCLUSIONS

The results of this study indicated that SCG and APR may be used as alternatives to conventional feed ingredients because of their nutrient composition and relatively low cost. Pre-fermentation of these products, however, may be inappropriate for improving their nutritional content considering the increase in production costs. Further studies on *in vivo* ruminal fermentation, and whole body digestion and metabolism by supplementing these by-products are warranted.

### Funding

This research was supported by a Bio-industry Technology Development Program (Project No. 312030-04-3-HD060), Ministry of Agriculture, Food and Rural Affairs. The funders had no role in study design, data collection and analysis, decision to publish, or preparation of the manuscript.

## Grant Disclosures

The following grant information was disclosed by the authors:

Bio-industry Technology Development Program: 312030-04-3-HD060.

Ministry of Agriculture, Food and Rural Affairs.

## Competing Interests

The authors declare there are no competing interests.

## Author Contributions

- Jakyeom Seo conceived and designed the experiments, performed the experiments, analyzed the data, contributed reagents/materials/analysis tools, wrote the paper, prepared figures and/or tables, reviewed drafts of the paper.
- Jae Keun Jung performed the experiments, reviewed drafts of the paper.
- Seongwon Seo conceived and designed the experiments, contributed reagents/materials/analysis tools, wrote the paper, reviewed drafts of the paper.

## Animal Ethics

The following information was supplied relating to ethical approvals (i.e., approving body and any reference numbers):

Chungnam National University Animal Research Ethics Committee —Approval number: CNU-00455.

## Supplemental Information

Supplemental information for this article can be found online at http://dx.doi.org/10.7717/peerj.1343#supplemental-information.

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
