# Peer review of "Evaluation of nutritional and economic feed values of spent coffee grounds and *Artemisia princeps* residues as a ruminant feed using *in vitro* ruminal fermentation"

_PeerJ, doi:10.7717/peerj.1343_

## Round 0.1 · original submission · Major Revisions

Dear Dr Seo,

Thank you for submitting your manuscript to PeerJ. Two expert reviewers have examined it and after careful consideration, we feel that it has merit but is not suitable for publication as it currently stands.

We invite you to submit a revised version that addresses the points raised.

Yours sincerely,
Maria Rosaria Corbo

Reviewer 1 ·

Basic reporting

See below

Experimental design

Material and methods
Line 133: the author should report the name of the buffer used to mix the rumen fluid

Validity of the findings

Introduction:
The author should revise the references and delete the oldest.
Line 77: the author should add “total mixed rations” besides its abbreviation in this line.

Results:
Line 196-200; table 2: the author should rewrite this passage to make it easier to match with the table.
Line 198 – 199- 206- 211-215: The author should clarify the unit of measure
Line 209: the use of the word trend is not justified; please, change the sentence (e.g: there was a decrement…) in order to avoid misunderstanding.
Line 212: the author should write “diet” instead of “treatments”

Discussion:
In my opinion, the discussion need to be revised and improved in the light of the table modifications written below and rewrite with more criticism.
Line 272: the author should delete the word “trend”

Table:
Table 1 and 2: the author should report the p value of the treatments
Table 3: the author should report the standard error of the mean

Additional comments

The major point of criticism was the discussion section. In my opinion, the discussion need to be deeply revised and improved.

I would recommend major revision of the paper before it can be considered for publication.

Reviewer 2 ·

Basic reporting

No comments

Experimental design

No comments

Validity of the findings

No comments

Additional comments

The effects of such feed supplements require further studies focused on the effects in terms of the nutritional and commercial quality of various food derivatives including meat and milk based products.

---

## Round 0.2 · accepted · Accept

Dear Dr. Seo,

I'm pleased to inform you that your paper in its revised form is now suitable for publication on PeerJ.

Best regards,
Maria Rosaria Corbo